# Seedlessness Trait and Genome Editing—A Review

**DOI:** 10.3390/ijms24065660

**Published:** 2023-03-16

**Authors:** Md Moniruzzaman, Ahmed G. Darwish, Ahmed Ismail, Ashraf El-kereamy, Violeta Tsolova, Islam El-Sharkawy

**Affiliations:** 1Center for Viticulture and Small Fruit Research, College of Agriculture and Food Sciences, Florida A&M University, Tallahassee, FL 32308, USA; 2Faculty of Agriculture, Department of Biochemistry, Minia University, Minia 61519, Egypt; 3Department of Botany and Plant Sciences, University of California Riverside, Riverside, CA 92521, USA; 4Department of Horticulture, Faculty of Agriculture, Damanhour University, Damanhour P.O. Box 22516, Egypt

**Keywords:** CRISPR/Cas, genome editing, molecular breeding, ovule abortion, parthenocarpy, seedlessness, stenospermocarpy

## Abstract

Parthenocarpy and stenospermocarpy are the two mechanisms underlying the seedless fruit set program. Seedless fruit occurs naturally and can be produced using hormone application, crossbreeding, or ploidy breeding. However, the two types of breeding are time-consuming and sometimes ineffective due to interspecies hybridization barriers or the absence of appropriate parental genotypes to use in the breeding process. The genetic engineering approach provides a better prospect, which can be explored based on an understanding of the genetic causes underlying the seedlessness trait. For instance, CRISPR/Cas is a comprehensive and precise technology. The prerequisite for using the strategy to induce seedlessness is identifying the crucial master gene or transcription factor liable for seed formation/development. In this review, we primarily explored the seedlessness mechanisms and identified the potential candidate genes underlying seed development. We also discussed the CRISPR/Cas-mediated genome editing approaches and their improvements.

## 1. Introduction

Seedlessness is one of the most valuable agricultural traits in fruit crops that consumers appreciate for fresh consumption and value-added processed products [1,2]. It enriches the eating quality of the fruits due to their expanded edible pulp and the absence of hard seeds with an awful taste. Further, seedlessness could prevent browning and bitterness caused by seeds [3]. Moreover, it improves many other fruit biometric characteristics regarding acid/sugar levels, dry matter, firmness, and overall shelf-life qualities of climacteric fruit due to reduced ethylene generated by seeds [4]. Seedlessness can also mitigate fruit yield losses caused by environmental stresses that affect pollination and fertilization processes [5]. Finally, it occurred independently of pollination and fertilization, which increases fruit production, particularly in dioecious species, due to the uselessness of the pollen source staminate trees. Studies on fruit seedlessness suggest that the trait is coordinated by intricate systems involving hormonal, genetic, and environmental factors [6,7]. Therefore, there are many causes underlying the seedless fruit set program [8]. The most classical reasons include male sterility, degradation of mother pollen cells, embryonic abortion, and chromosomal irregularities during meiosis leading to triploidy.

In typical seeded fruit, the ovary proliferates after fertilization through a coordinated program of molecular, biochemical, and structural changes that stimulate fruit size enlargement due to the interplay of cell division, differentiation, and expansion of sporophytic and gametophytic tissues [9]. Research on mechanisms underlying fruit seedlessness has highlighted the potential involvement of two distinct strategies, parthenocarpy and stenospermocarpy [10]. In parthenocarpy, true seedlessness occurs, and the ovary develops into fruit independent of pollination and fertilization [11]. However, two different procedures were identified for the parthenocarpic fruit set program. The obligatory-parthenocarpy, where a plant always produces seedless fruits (i.e., pineapple), and the facultative-parthenocarpy, where seedless fruits only develop if pollination is prevented (i.e., watermelon) [12]. Parthenocarpic fruit development is triggered by the deregulation of the hormone balance in ovary tissues, mainly auxin, gibberellins (GAs), and/or cytokinins (CKs). Applying these hormones to unpollinated ovaries at anthesis can stimulate pollination-independent ovary growth and produce parthenocarpic seedless fruit, strongly supporting their individual and overlapped roles during early fruit development [13]. An earlier study reported that the growth of tomato fruit is coordinated by a delicate balance between auxin and GA, whereby auxin is needed to mediate cell division and GA is required to organize cell expansion [7]. The parthenocarpy trait stability in fruit crops primarily occurs through elective pressure for seedlessness during domestication and breeding [8]. However, parthenocarpic genotypes were also identified in wild species and non-fruit crops [14].

In stenospermocarpy, pollination and fertilization typically occur; however, the seed growth is prematurely aborted due to the cessation of seed coat and endosperm development, resulting in expanded fruit size with seminal rudiments or seed traces [15]. The fact that different degrees of seedlessness were observed in progeny grapevines resulting from crossing seeded and stenospermocarpy seedless parents adds more complexity to the integrative regulatory network and signaling pathways underlying the stenospermocarpy seedless fruit set machinery [16]. Despite recent advances in grape biology, the molecular basis that triggers stenospermocarpy fruit development is largely unknown [10]. The efforts to unravel the molecular basis for stenospermocarpy in grapes were able to identify and functionally characterize several genes that can be potentially involved in the procedure [17,18,19]. Although the results did not show an ultimate gene network, they at least shed light on potential molecular mechanisms that synchronize stenospermocarpy machinery.

Fruit size and weight are positive commercial attributes, through which the number of developed seeds per fruit is positively correlated with the two characters [20]. Parthenocarpy fruit set results in considerably smaller fruit size than seeded fruit due to the absence of seed initiation, leading to reduced hormone levels necessary to sustain fruit growth [7]. However, stenospermocarpy does not compromise or, in the worst-case scenario, slightly reduce the fruit size because the ovary-growth event occurs after pollination and fertilization, making stenospermocarpy seedlessness a more attractive trait for breeding (Figure 1). CKs are essential to determining ovary size before fertilization. However, the slightly compromised size of the stenospermocarpy fruit is due to the availability of CKs post-fertilization, which negatively regulate cell expansion during fruit development [21].

Parthenocarpy seedlessness can be induced by applying hormones to unpollinated inflorescences at anthesis, via fostering self-incompatibility, or through generating triploid plants using conventional breeding practices [4,22]. Nevertheless, all the strategies are laborious, time-consuming, and sometimes not possible to use due to the absence of proper parental genetic resources. In the meantime, no treatment or application that can induce stenospermocarpy seedlessness has been identified yet. Accordingly, both seedlessness mechanisms are important, depending on growth conditions and commercial value. This has opened up the opportunity for the genetic engineering approaches that have given encouraging results, both in the quality and quantity of seedless fruit production.

Genetic engineering technology is a promising approach that has contributed considerably to crop improvement. Over 20 genetically modified (GM) crop entities had been commercialized by 2018—including soybeans, cotton, maize, and canola—with a share of world production ranging between 29–78% [23,24]. Some other GM crops are produced depending on the country, such as potato, apple, alfalfa (North America), papaya (Hawaii), eggplant, squash, safflower, pineapple, and sugar cane (different countries) [25]. The leading countries for GM crop cultivation are the USA, Brazil, Argentina, Canada, and India, with total productive land spaces ranging between 11.6–75 M ha, in addition to Paraguay, China, Pakistan, South Africa, and Australia, with entire land spaces ranging between 0.8–5 M ha [23,26]. Before commercialization, these crops had to pass through a prolonged and extensive regulatory process. The US Department of Agriculture (USDA) does not impose any GMO regulations on plants with targeted mutagenesis by self-repair mechanisms if they are free from *Agrobacterium*, any transgene, or foreign genetic materials. Accordingly, there is a high probability that CRISPR/Cas RNPs could be exempt from current GMO regulation [27,28]. CRISPR/Cas technology has been implemented to gain desired traits in many crops [29,30]. For instance, the disease resistance trait was developed in rice [31], tomato [32,33], cucumber [34], wheat [35,36], citrus [37], and *Arabidopsis* [38] by engineering disease-susceptible genes. However, the technology needs a comprehensive understanding of the gene(s) coordinating the desired trait. Accordingly, several other factors are essential, such as appropriate gRNA selection, promoter choice, and a suitable Cas protein. In this review article, we investigate and scrutinize genes that function as positive regulators of seed formation. Further, we discussed diverse CRISPR/Cas genome editing approaches to introduce a seedless character. The knowledge and information generated from this review will improve our critical thinking toward introducing innovative, high-value quality attributes to fruit crops.

## 2. Genes Coordinating the Seedlessness Trait

### 2.1. Auxin-Related Genes

Applying synthetic auxins to unpollinated flowers induces parthenocarpic fruit growth by modulating early cell division, resulting in an increase in the pericarp volume [4]. Several molecular studies have demonstrated the role of auxin in triggering and coordinating the transition from flower to fruit [39,40,41,42]. The auxin-mediated parthenocarpy seedless induction occurs by either altering auxin synthesis or signaling.

In the auxin pathway, seven gene candidates acting as positive regulators of the seeded fruit set program were identified (Figure 2; Table 1). The *SlIAA9* and *SlARF7*/*AtARF8* TFs belong to the AUX/IAA and auxin response factor (ARF) gene families, respectively.

Aux/IAAs and ARF-transcriptional regulators interact in homo- and heterodimers, forming complexes that repress auxin-dependent changes in gene expression and, therefore, auxin action. Auxin binding to an F-box receptor promotes SCF^TIR1/AFB^ complex formation, leading to the ubiquitin-dependent proteolysis of Aux/IAA [43,44]. Loss of Aux/IAA repressors allows ARF-mediated auxin-responsive changes in gene transcription. Transgenic tomato and *Arabidopsis* plants with suppressed *SlIAA9*, *SlARF7*, or *AtARF8* mRNA displayed fruit/silique development before fertilization, giving rise to parthenocarpy seedless fruit development [45,46]. Similarly, the amiRNA *SlARF5* lines exhibited ovary growth and formed seedless tomato fruits following emasculation. These parthenocarpic fruits developed fewer locular tissues, and the fruit size and weight declined in transgenic lines compared to wild-type fruits [47].

The transcriptional co-repressor (TPL) is an upstream central regulatory hub to control phytohormone pathways. SlTPLs participate in the auxin-signaling pathway by interacting with Aux/IAA proteins in tomatoes, particularly IAA9. There is no interaction between SlTPL1 and the ARF activators ARF7, ARF8, or ARF5. Accordingly, IAA9 is the connection link between SlTPL1 and ARFs. The down-regulation of *SlTPL1* in tomato plants produced facultative parthenocarpy fruit associated with a significant decline in the expression of ARF-related genes. Transgenic *SlTPL1*-RNAi plants produced WT-like fruits having no pleiotropic effect under normal growth conditions [48].

Another gene, *PARENTAL ADVICE-1* (*Pad-1*), encodes an aminotransferase, which is involved in auxin homeostasis. The role of Pad-1 in unpollinated ovaries is to prevent the excessive accumulation of IAA, resulting in a precocious fruit set. A loss-of-function mutant, *pad-1* caused high accumulation of IAA in the tomato and pepper ovaries, suggesting that Pad-1 protein is involved in auxin homeostasis during ovary development [49].

Similarly, several genes involved in auxin transport were identified as master genes that enhance parthenocarpy fruit set. The *AUxin Cum Silencing Action* (*AUCSIA*) is a green plant gene family encoding a mini-protein involved in several aspects of auxin biology, including polar auxin transport [50,51]. Silencing of the *AUCSIA* in tomato and *Arabidopsis* caused fruit set independent of pollination and fertilization that produced facultative and obligatory parthenocarpic fruits [51]. Further, auxin efflux transport is conducted by the PIN-FORMED (PIN) protein family. In tomato, the application of auxin efflux transport inhibitors produced parthenocarpic fruit development. Silencing of the *SlPIN4* gene resulted in parthenocarpic fruits due to precocious fruit development [52]. Finally, *chalcone synthase* (*CHS*) encodes a key enzyme that catalyzes the first committed step in the flavonoid biosynthetic pathway [53]. Flavonoids act as negative regulators of auxin transport that affect auxin sensitivity [54]. Down-regulation of *CHS* mRNA produced parthenocarpic seedless tomato fruits probably by enhancing polar auxin transport [55].

### 2.2. Gibberellin-Related Genes

Applying active gibberellins (GA_1_, GA_3_, or GA_4_) to unpollinated flowers induces parthenocarpy fruit set in several plant species [39,41]. The role of GAs in the fruit set was also supported by the analysis of the natural tomato mutants (pat, pat2, and pat-K) that produce parthenocarpic fruit [56,57]. In higher plants, it is essential to maintain optimal levels of phytohormones to ensure typical growth and development (Figure 2). Hence, plants have to retain a mechanism to remove any excess active compounds or their biosynthetic precursors to ensure the proper function of phytohormones. Such a strategy can prevent the progressive accumulation of hormones. The flux of active GAs is regulated by the balance between their rates of biosynthesis and deactivation. The GA20ox and GA3ox genes encode key enzymes of bioactive GA synthesis, whereas GA2ox is the major GA inactivation enzyme [58]. Modifying the regulation of genes by adjusting GA flux can subsequently alter the processes regulated by GA [59]. *GA2oxs*-silenced tomato plants displayed parthenocarpic fruit growth. However, the mutant plants exhibited branching inhibition due to the high accumulation of active GA_4_ in axillary buds [60].

Seedless fruits have also been produced by modifying the GA-signaling pathway. According to the relief of restraint model, DELLA proteins operate as growth repressors, and GA-mediated DELLA degradation is a critical step to overcome this restraint [61]. At low GA levels, DELLA proteins impair the activity of basic helix-loop-helix (bHLH) transcription factors by interacting with their DNA binding domain [62]. The binding of GA to its GID1 receptor results in a conformational change that promotes the interaction of GID1 with DELLA [63]. The GA–GID1–DELLA complex is subsequently recognized by the SCF^SLY1/GID2^ E3 ubiquitin-ligase complex, which mediates the ubiquitination of DELLA proteins. This ubiquitin mark destines the DELLA proteins for degradation via the 26S proteasome, thereby allowing growth by releasing their inhibitory interaction with GA-dependent gene partners. In agreement with their function as growth repressors, lacking one or more DELLA proteins within the plant elicited constitutive activation of the GA-signaling pathway independent of GA presence, in which the mutant plants exhibited a GA-overdose phenotype, including parthenocarpic fruit development [64]. Antisense *DELLA* tomato and *Arabidopsis* plants produced seedless fruits [65]. However, the resultant fruits were smaller and displayed elongated shapes compared with typical fruits.

### 2.3. Cytokinin-Related Genes

CKs influence seed development and play other roles in plant growth [39]. They are involved in regulating ovary size and ovule development, which affect seed size and number [21]. Polycomb group (PcG) proteins regulate the expression of the signature CKs genes [66]. In *Arabidopsis*, the genes of *MEDEA* (*MEA*), *FERTILIZATION INDEPENDENT ENDOSPERM* (*FIE*), and *FERTILIZATION INDEPENDENT SEED 2* (*FIS2*) encode the PcG protein that controls seed development via synchronizing embryo and endosperm proliferation (Figure 2). The MEA protein holds a characteristic SET domain that confers histone methyltransferase activity. The *mea* mutation caused seed abortion in *Arabidopsis*, primarily mediated by epigenetically deregulating the expression of the type I MADS-box gene *PHERES1* (*PHE1*) [67].

The *FIE* gene directly contributes to female reproductive development. The mutant *fie* by the female gametophyte caused embryo abortion by influencing the central cell’s development. Mutants of *fie* can replicate the central cell nucleus and stimulate endosperm development independent of fertilization procedures [68]. Similarly, *fis1* and *fis2* mutants generated barely-formed pro-embryos that did not develop beyond the globular stage, causing seed abortion. The arrested embryos’ emerged due to continued endosperm growth until the cellularization stage [69]. Further, *MULTICOPY SUPPRESSOR OF IRA 1* (*MSI1*) is a WD-40 domain protein that forms a complex with the MEA and FIE proteins. It is another member of the conserved *FIS* polycomb group complex that belongs to the PRC2 type. Mutant plants heterozygous for *msi1* were able to produce parthenocarpic siliques in *Arabidopsis* [70]. The *msi1* mutant gametophytes initiated endosperm development in the absence of fertilization at high penetrance. It showed a seed abortion ratio of 50%, with seeds aborting when the mutant allele is maternally inherited, irrespective of a paternal WT or mutant *MSI1* allele.

*DNA methyltransferase 1* (*MET1*) is a central regulator of parentally imprinted genes that affect seed growth. The *met1* loss-of-function mutant caused a reduction in seed size, presumably linked to the silencing of the paternal allele of growth enhancers in the endosperm, which nurtures the embryo [71]. *MET1* and *MEDEA* exhibited overlapping expression patterns in reproductive tissues pre- and post-fertilization. Apparently, there is a mechanistic association between two major epigenetic pathways involved in histone and DNA methylation in plants through the physical interaction of MET1 with FIS-PRC2, the core component of MEDEA. This concerted action is relevant for the repression of seed development in the absence of fertilization [72].

### 2.4. MADS-Box Genes

The MADS-box gene family of transcription factors (TFs) are crucial regulatory networks underlying multiple developmental pathways in plants, animals, and fungi [73,74]. MADS-box proteins interact with members of the same family or with diverse other proteins to orchestrate different developmental programs that respond to external and internal stimuli signals, such as growth-, hormone-, and defense-signaling [75]. Based on structural characteristics, the MADS-box TFs are classified into two major groups: type I and type II [76]. Type I MADS-box TFs hold an SRF-like domain, whereas type II comprises the Myocyte Enhancer Factor 2-like (MEF2-like) domain, known as the MIKC genes in plants [77]. MIKC genes can be further divided into MIKC^C^ and MIKC* subfamilies. MIKC^C^ genes have been widely reported due to their involvement in diverse biological functions in plants, particularly floral organ specification, flowering time regulation, and fruit development and ripening [78,79]. The approach used to elucidate MADS-box gene function was through the analysis of plant phenotypes resulting from their downregulation or overexpression.

Several lines of evidence have associated seed initiation/development and the seedless fruit set program with the alteration of several MADS-box members belonging to type II lineage MICK^C^ subfamilies (Figure 2). The silencing of different tomato MADS-box gene members of class B, including *TAP3*, *TM6*, *SlGLO1*, and *SlGLO2*, produced mutant plants that developed fruits with no or few seeds [80]. For instance, the downregulation of the tomato floral homeotic gene *APETALA3* (*TAP3*) in the ovary results in male sterility and parthenocarpic fruit development [81]. Emasculation and manual pollination assays using WT pollen suggested a liable pollen impairment phenotype involved in the facultative parthenocarpy fruit set of the *TAP3*-silenced plants [82]. The parthenocarpic fruit development in *TAP3*-downregulated ovaries was associated with increased GA levels, suggesting that stamen development negatively regulates fruit set by repressing GA biosynthesis. A different floral homeotic gene that belongs to the same family, designated as *PISTILLATA* (*MdPI*), has been identified as the determinant underlying parthenocarpy fruit set in apples [83]. In addition to the altered fruit set program, the apple mutant trees produce a distinct flower phenotype, where petals are converted to sepals and stamens to carpels. The apple mutants exhibited retrotransposon insertion events in intron 4 or intron 6, which abolished the typical *MdPI* gene expression.

Another distinct tomato gene, *AGAMOUS-Like 6* (*TAGL6*), encodes a MADS-box protein of the subfamily AGL6 [78]. Transcriptome analysis followed by marker-assisted mapping established that a mutation in *TAGL6* is responsible for an interesting EMS-induced tomato mutant [84]. The mutant plants exhibited a facultative parthenocarpy fruit phenotype under heat stress conditions without pleiotropic effects on vegetative and reproductive development. The *Tagl6* mutation showed typical characteristics of WT plants, excluding the parthenocarpic fruit set, in terms of pollen viability, sexual reproduction capacity, and fruit biometrics, making *TAGL6* an attractive target gene for facultative parthenocarpy. Gene expression analysis and CRISPR/Cas9 gene knockout confirmed the role of *TAGL6* as a critical regulator coordinating the transition from the state of ‘ovary arrest’ to fertilization-triggered fruit set resumption. Once the down-regulation of *TAGL6* is alleviated, the ovary/fruit development resumes and continues to reach its full potential.

The *Tomato MADS-box 29* (*TM29*) gene is the ortholog of the *Arabidopsis SEPALLATA* (*AtSEP*) genes that belong to the E class [78]. Based on transcript abundance and evaluation of silenced mutants, it was suggested that *TM29* behaves like *AtSEP1* via coordinating floral organ development and identity. The *tm29* plants produced aberrant flowers with phenotypic alterations in the organs of the inner three whorls [85]. The yellow petals and stamens have been converted into green color. The reproductive organs of stamens and ovaries were sterile; however, the ovaries continued growing into parthenocarpic fruit. The fruits were malformed, as they emerged from ectopic shoots with partially developed leaves and secondary flowers. These shoots resembled the primary transgenic flowers and continued to produce parthenocarpic fruit and ectopic shoots.

Finally, an interesting homeotic MADS-box gene classified as a D member *AGAMOUS-Like 11* (*AGL11*) was demonstrated to coordinate seed development. Several phenomic, genetic, biochemical, and transcriptomic approaches allowed the identification of the *Arabidopsis AGL11* gene, called *SEEDSTICK* (*STK*), as a master regulator coordinating ovule identity and the flavonoid pathway, particularly proanthocyanidins synthesis linked to seed coat development [86,87]. Afterward, an *AGL11* ortholog named *SHELL* was identified in oil palm as a crucial regulator for the thickness of the coconut-like shell surrounding the kernel [88]. Through a homozygosity mapping by sequencing approach, it was demonstrated that the thin kernel shell phenotype was associated with two independent mutations within the DNA-binding domain. Later, *VviAGL11* was identified as a substantial gene controlling seed morphogenesis in cultivated grapevine [16,18]. A missense mutation detected within the C-terminal region of the gene was associated with a reduced *VviAGL11* transcription level and subsequently considered the direct cause of triggering the seedless stenospermocarpy fruit set program. Interestingly, the genetic characterization of the mutant highlighted the dominant inheritance of the seedless trait [16]. Subsequently, two *AGL11* homologs were identified in tomato, *SlAGL11* and *SlMBP3* [89]. Genetic analysis of numerous tomato genotypes, along with the functional analysis of the two genes via CRISPR/Cas9 and silencing approaches, suggested the critical role played by *SlMBP3* in regulating the structure of locular tissue in tomatoes. Individual knockout mutations did not influence seed development; however, the dual *SlMBP3*/*SlAGL11* mutant lines displayed smaller plants with a dramatic reduction in fruit size/weight and under-developed seeds showing a complete inability to germinate. Apparently, SlMBP3 and SlAGL11 have overlapping functions in seed development, through which the absence of an ortholog can be covered by the presence of the other.

### 2.5. Other Genes

#### 2.5.1. HYDRA (HYD) Gene

The *SPOROCYTELESS/NOZZLE* (*SPL/NZZ*) gene is a floral organ-building gene that encodes a protein related to MADS-box transcription factors. The *SPL/NZZ* plays a central role in controlling early anther cell differentiation and stamen identity [90,91,92]. The direct activation of SPL/NZZ by the MADS-box AG is instructed for early anther development [93]. The tomato *HYDRA* gene (*HYD*) encodes a putative SPL/NZZ transcription factor. The *SlHYD* is essential for preventing precocious ovary growth, flower maturation, and an appropriate fruit set program [94]. The tomato *hyd* mutant produces seedless fruit due to the impaired formation of male and female germlines, triggering parthenocarpic fruit set development. Interestingly, the precocious growth of the ovary in the *hyd* mutant was associated with changes in the expression of genes involved in gibberellin (GA) metabolism, particularly the accumulation of *SlGA3ox* and the suppression of *SlGA2ox* (Figure 2).

#### 2.5.2. Binding Cassette G Transporter

The ATP-binding cassette (ABC) transporter is one of the largest and oldest protein families. The tonoplast-localized ATP-binding cassette pumps various secondary metabolite substrates across the vacuolar membrane into the vacuole using the energy generated by ATP [95]. The *VviABCG20* encodes a putative ATP-binding cassette G transporter in grape. The gene was identified as a differentially expressed gene during seed development or seed abortion of the seeded and stenospermocarpy seedless grapes, respectively (Figure 2) [19]. Silencing of the *VviABCG20* ortholog in tomato (*SlABCG20*) resulted in plants that set fruit with no or few seeds, suggesting its potential involvement in seed development. Interestingly, the *VviDof14* gene, which acts as a negative regulator of *VviABCG20*, showed a higher expression level in Thompson seedless grape [96].

**Table 1 ijms-24-05660-t001:** Transcription factor genes involved in seed formation could be utilized to induce seedlessness.

Gene Name	Species	Protein	References
**Auxin-related genes**			
IAA9	*S. lycopersicum*	Auxin repressor Aux/IAA 9	[40]
ARF7/8	*A. thaliana*; *S. lycopersicum*	Auxin-response factor 7/8	[45,46]
ARF5	*S. lycopersicum*	Auxin-response factor 5	[47]
TPL1	*S. lycopersicum*	Transcriptional co-repressor TOPLESS 1	[48]
Pad-1	*S. lycopersicum*	Proteasome subunit alpha type-7	[49]
AUCSIA	*S. lycopersicum*	AUxin Cum Silencing Action	[51]
PIN4	*S. lycopersicum*	Auxin efflux carrier component 4	[52]
CHS	*S. lycopersicum*	Chalcone synthase	[55]
**Gibberellin-related genes**			
GA2ox	*S. lycopersicum*	Gibberellin 2-oxidase	[60]
DELLA	*A. thaliana*; *S. lycopersicum*	DELLA protein GAI	[64,65]
**Cytokinin-related genes**			
MEA	*A. thaliana*	SET domain-containing protein	[67]
FIE	*A. thaliana*	Transducin/WD40 repeat-like superfamily protein	[68]
FIS2	*A. thaliana*	VEFS-Box of polycomb protein	[69]
MSI	*A. thaliana*	Transducin/WD40 repeat-like superfamily protein	[70]
MET1	*A. thaliana*	DNA (cytosine-5)-methyltransferase 1	[71,72]
**MADS-box genes**			
TAP3	*S. lycopersicum*	Tomato APETALA 3	[81]
PI	*Malus domestica*	PISTILLATA	[83]
TAGL6	*S. lycopersicum*	AGAMOUS-Like 6	[84]
TM29	*S. lycopersicum*	Tomato MADS-box 29	[85]
AGL11	*V. vinifera*; *E. guineensis*	AGAMOUS-Like 11	[16,88]
**Other genes**			
HYDRA	*S. lycopersicum*	SPOROCYTELESS/NOOZLE-like protein	[94]
ABCG20	*A. thaliana*	ABC-2 type transporter family protein	[96]

## 3. Genome Editing Technology—CRISPR-Cas

Targeted genes or genome editing technologies have been explored for the last three decades. The zinc finger nucleases (ZFNs), transcription activator-like effector (TALE) nucleases (TALENs), and clustered regularly interspaced short palindromic repeats (CRISPR) are the three primary strategies that have been developed and utilized for genome editing. However, the CRISPR-associated protein 9 (CRISPR-Cas9) technology has emerged as the most potent tool [97]. In general, CRISPR-mediated genome editing mandates several requirements, including a guide RNA (gRNA) composed of 20 synthetic nucleotide sequences that binds to target DNA and a nuclease enzyme (Cas9) that breaks the DNA near the protospacer adjacent motif (PAM) sequence [98]. Cas9 is not active under natural conditions, as the active Cas9-sgRNA complex can only be assembled when the Cas9 encloses the sgRNA. Then, the active complex scans the double-strand DNA to identify and bind to the complementary sequences. Afterward, the enzyme’s HNH domain cleaves the DNA closely before the PAM sequence, while the RuvC domain breaks another strand, causing a double-strand break (DSB). The DSBs are eventually repaired by endogenous DNA repair mechanisms, such as non-homologous end joining (NHEJ) and homology-directed repair (HDR), causing nucleotide insertions and/or deletions (indels) at the desired sites.

## 4. Improved CRISPR-Cas9 Technology

A notable deficiency of the early experiments with the CRISPR/Cas9 system was the high rate of off-target cleavages caused by the formation of a mismatched complex between gRNA and DNA. Several strategies were reported to improve target site specificity and efficiency, such as modifying the Cas9 enzyme [99,100], increasing the length of the PAM sequence [101,102,103], generating new Cas proteins (i.e., CRISPR-Cas12a) [104], and modifying the CRISPR technology itself.

Several changes were applied to the Cas9 enzyme via modifying the cleavage domain of Cas9-D10A or Cas9-H840A, which enriched the specificity of cleaving target DNA [99,100]. The VQR variants of SpCas9 recognize NGA PAMs, and the VRER variants recognize NGCG PAMs, greatly expanding the genome editing range [101]. Moreover, Cas9 can be deactivated (dCas9 or CRISPRi) by a point mutation in the RuvC and HNH nuclease domains. Co-expression of dCas9 and a sgRNA prevents transcription elongation and subsequently averts protein function, which diminishes gene expression [105]. Gene expression can also be regulated by fusing dCas9 with a repressor or an activator. For instance, dCas9-VP64 and dCas9-p65AD can efficiently trigger gene expression [106]. Cas9 combined with histone-modified/DNA-methylated enzymes can modulate the epigenetic modification of genes [107]. A light-controlled endogenous gene expression circuit was reported. This circuit was developed by fusing the light-inducible proteins (CRY2 and CIB1) to a transactivation domain and dCas9 [108]. Furthermore, the fusion of Cas9 with a fluorescent protein was declared to label the DNA in a particular compartment, facilitating the study of complex chromosomal architecture and nuclear organization [109].

The discovery of different nucleases that recognize different PAMs has enriched the CRISPR/Cas strategy in terms of specificity and efficiency. For instance, the Nmecas9 enzyme, derived from *Neisseria meningitidis*, recognizes an 8-mer (50-NNNNGATT) PAM. This longer PAM sequence can reduce off-target cleavage and increase target specificity [110]. Likewise, the Sacas9 enzyme, derived from *Staphylococcus aureus*, recognizes a 6-mer (50-NNGRRT) PAM sequence [99]. The St1cas9 and St3cas9 derived from *Streptococcus thermophilus* recognize a 7-mer (5′-NNAGAAW) and a 5-mer (5′-NGGNG) PAM, respectively. The St1cas9 and St3cas9 minimized off-target rates while editing the human PRKDC and CARD11 loci, compared to the enzyme SpCas9 derived from *Streptococcus pyogenes* [102]. Furthermore, CRISPR-CpfI, developed from *Prevotella* and *Francisella*, is a class II type V endonuclease that recognizes the 5′-TTTN-3′ PAM sequence [111]. It can be used effectively in plants and animals with reduced or no off-target effects [112]. Unlike Cas9, another endonuclease, NgAgo derives from *Natronobacterium gregoryi* and operates on 24 nucleotides of ssDNA with 5′-phosphorylation as a guide. It can bind 5′-phosphorylated single-stranded guide DNA (gDNA) of ~24 nucleotides and efficiently generate gDNA sequence-specific DNA double-strand breaks. This editing process does not require the presence of PAM [113]. Finally, the C2c2, derived from *Leptotrichia shahii*, showed two nuclease functions that cleave single-stranded RNA [114].

In addition to the previous efforts to improve Cas9 engineering, several studies were performed to enhance the CRISPR technology. Highly efficient multiplex genome editing showed the new dimensions of plant biology and crop breeding. Sophisticated genetic engineering objectives became feasible, including multigene knockouts, gene or promoter knock-ins, gene activation and repression, chromosomal deletion and translocation, epigenome modifications, and many others. Multiplex editing was executed by expressing Cas9 (or Cas9-derived effectors) together with multiple gRNAs targeting multiple sites. The multiple gRNA toolbox system can be constructed by either expressing the gRNAs individually or simultaneously. The two strategies have advantages and disadvantages regarding cloning readiness, the number of expressed gRNAs, and editing efficiency.

Moreover, the recently developed CRISPR-TSKO technology can evaluate gene function based on a tissue-specific knockout [115]. CRISPR-TSKO is able to modify the genome of specific cells, tissues, and organs of different allelic backgrounds for plant disease-resistant capacity engineering. This tissue-specific knockdown can be a better option for comprehensively understanding signaling and tolerance mechanisms. Cas9 expression under the control of the egg cell-specific promoter EC1.2 and the germline-specific promoter SPL produced a heritable mutant in *Arabidopsis* [116,117]. Recently, Cas9 was expressed under the control of the fiber-specific -*NST3/SND1* promoter to target the essential *Arabidopsis* gene *HCL* (encoding a hydroxycinnamoyl transferase) [118].

Base editing (BE) is another newly developed strategy for precise genome editing that enables irreversible base conversion at a specific site. The BE machinery is a complex of a catalytically impaired Cas protein, guide RNA (gRNA), and nucleobase deaminase domain that can convert specific base pairs [119]. All four transition mutations, C → T, G → A, A → G, and T → C, can be introduced in the genome with the available CRISPR/Cas base editors. The cytosine base editor (CBE) can establish a G–C to A–T mutation, while the adenine base editor (ABE) can alter an A–T base pair into a G–C. In RNA, the conversion of adenine (A) to inosine (I) is also possible with the RNA base editor [120].

The latest addition, “Prime Editing” is based on “search-and-replace”. The technology can force targeted insertions/deletions within the gene. Interestingly, it does not require double-strand breaks (DSBs) or donor DNA templates. The earlier version of prime editors (PE1) uses RNA-programmable nickase and a prime editing guide RNA (pegRNA) fused with reverse transcriptase (RT). The updated PE2 version exhibited higher editing efficiency because it used an engineered RT. While the latest PE3 version utilizes two guide RNAs and further increases the editing efficiency by producing nick at specific locations on the non-edited strands to induce its replacement. Prime editing offers much lower off-target activity than Cas9, far fewer byproducts with similar/higher efficiency than Cas9-initiated HDR, and complementary strengths over base editor technology [121].

## 5. Genetic Engineering Strategies for Seedlessness Breeding

Parthenocarpy seedless fruits can be accomplished either by exogenous application of plant growth regulators, conventional breeding, interspecies hybridization, or polyploidy breeding. However, none of these strategies is feasible to induce stenospermocarpy seedless. For instance, muscadine seedless grape breeding is not viable due to the absence of the trait within the species. The only available seedless muscadine genotype, “Fry Seedless”, is parthenocarpic with limited commercial value and cannot be used as a crossing parent in the breeding program due to male sterility (Figure 3) [122]. Muscadine and bunch grape are classified under the *Euvitis* genera; however, the pronounced differences in their phenomic, metabolomic, and genomic characteristics represented by the dissimilarities in stress responses, horticultural and reproductive growth characteristics, and genome structure enabled us to classify them into two different genera, *Muscadinia* and *Vitis* [123]. Accordingly, introducing the stenospermocarpy seedless trait to muscadine grapes via generating *Vitis* x *Muscadinia* interspecific hybrids is challenging due to the differences in chromosome number and genetic incompatibility [124]. Alternatively, developing triploid seedless muscadine grapes might be an option that avoids the genetic barrier between species. However, the attempt to establish a triploid seedless muscadine grape did not produce satisfactory genotypes that can be promoted into new cultivars due to limited reproductive growth qualities [125]. Hence, genetic engineering could be a promising strategy for introducing a seedless trait.

Advanced genome-editing tools, such as CRISPR-TSKO, precise base editing, or prime editing approaches, were efficiently applied in different crops for precise genome editing that prevented or minimized the pleotropic effects [115,126]. However, this approach is considered GMO and may affect consumer acceptance. Interestingly, the Cas9-free lines can be selected by crossing out the transgenes from the segregating populations or through RNP-mediated protoplast transformation and regeneration [38,127]. Moreover, the CRISPR reagent (RNP) can be delivered to the germline cells using viral vectors like the Tobacco rattle virus (TRV). Thus, an inherited mutation could be achieved using the seeds from the germline-edited plant [128].

The model presented in Figure 4 illustrates our suggested strategy for introducing a seedless trait using genome editing technology.

**(I) Establishing a plant regeneration system based on embryonic suspension cell culture.** Genetic transformation using embryogenic cell suspension cultures is a good opportunity because of their higher organogenetic potential [129,130,131,132]. Regeneration of putatively transformed cells and subsequent grafting of transgenic micro-shoots on rootstocks may shorten the juvenile period for flowering and fruiting [133].

**(II) Selecting appropriate candidate gene(s) and regulatory elements for targeted genome editing.** The *AGL11* is considered the only identified upstream regulatory gene that controls the ovule/seed development, and its *Arabidopsis* mutant phenotype (*stk*) displayed compromised seed characteristics. [87]. In the case of *V. vinifera*, stenospermocarpy seedlessness is associated with a SNP mutation in *VviAGL11* (R197L) [16]. The availability of whole genome sequence (i.e., muscadine whole genome sequence) [134] facilitates target gene selection. In the current review, we highlighted many other genes associated with seed development/abortion (Table 1). These genes could be target candidates for genome editing. Organ-specific promoter-driven Cas protein expression has been reported on the CRISPR platform [116,135,136,137]. Using a seed-specific promoter could achieve the goal more effectively and efficiently because the desired expression will occur only in seeds. Seed development-specific promoters have been characterized using various genes and different plant species [96,138,139,140].

**(III) Cloning and assembly of a binary vector.** Guide RNA (gRNA) design for the particular gene and cloning the gRNAs into appropriate vector backbones are the primary tasks for genome editing vector construction. Different online tools for gRNA design (i.e., CRISPOR [141], CRISPR-P [142], CCTop [143], CHOPCHOP [144], and GuideMaker [145]) with customized features are openly accessible. High-efficiency cloning technology (MoClo) [146] and readymade cloning materials of different expression modules are readily available from addgene (https://www.addgene.org/) and other sources. Adopting the MoClo cloning technology would accelerate vector construction efficiency. Recently, it has been shown that gRNA possessing the same restriction site as the type II restriction enzyme used for the GG reaction does not affect MoClo cloning and subsequent genome editing efficiency, which expanded the gRNA selection options [147].

**(IV) Transformation of embryogenic cells or protoplasts and regeneration of putatively transformed cells into a complete plant.** Using the appropriate transformation system increases the chances of getting transformed events. Among the different types of transformation processes, *Agrobacterium*-mediated or direct protoplast transformation could be adopted. Interestingly, RNP (ribonucleoprotein) mediated genome editing of protoplasts could avoid current GMO regulations, as the USDA does not consider the plant a GMO if the engineering involves a plant self-repair mechanism.

**(V) Screening for potential transgenic events and securing approval from regulatory agencies.** The procedure is associated with both molecular and phenotypic evaluations. Molecular screening means genomic and proteomic studies of the desired genome engineering plant(s) to confirm that desired change(s) in the genome, and phenotypic screening means the study of visual changes (either positive or negative) in the plants. If satisfactory performance is achieved, the new plant genotype needs approval from regulatory authorities before release for commercialization.

## 6. Bio-Engineered Food Regulation and Consumer Acceptance

Conventional breeding has limited application for developing innovative value-added cultivars because of extreme heterozygosity, which is fostered by inbreeding depression. Perennial crops are characterized by long juvenility, extended breeding cycles, large plant size, poor fecundity, and high heterozygosity due to outcrossing fertilization. Accordingly, the development and introduction of improved cultivars is challenging, and the breeders lack the capacity to generate new cultivars quickly in response to evolving consumer/industry preferences and crisis circumstances (i.e., climate change). New cost-effective breeding technologies with obvious potential for enhanced improvement of economically viable crops have emerged from advances in genomic research and the refinement of cell culture tools. The technologies are well adapted to enrich precision breeding efficiency by enabling accurate, targeted, and reliable changes to the genome. This caused rapid changes in the landscape of life sciences, providing many novel biological applications by targeting several economically important traits. CRISPR/Cas technology has been developed to disrupt specific genomic loci with a very limited number of off-target alterations, resulting in plants with edited alleles. This method of delivery is effective in producing non-GMO plants due to its ability to avoid issues that arise from the stable insertion of T-DNAs into the genome.

Safety subject is a concern for bioengineered food. Accordingly, different regulatory organizations work in cooperation. Three U.S. organizations, including the food and drug administration (FDA), the U.S. department of agriculture (USDA), and the environmental protection agency (EPA) co-regulate the pre- and post-release of bio-engineered food products. In European Union (EU) countries, European food safety authority (EFSA) looks after the bioengineered products. The factor of evaluation between the U.S. and EU is also different. The U.S. approach focuses on the end product. Bop-engineered foods fall under the FDA classification of “generally recognized as safe”. They do not have to be approved before entering the market, and they typically do not require special labeling. However, the FDA recommends that companies go through a voluntary consultation process to determine whether their new GM foods would require premarket approval. Approval is necessary if the GM food contains high levels of toxic substances, allergens, or reduced levels of key nutrients. Interestingly, bioengineered food is not considered GM if the product is free from *Agrobacterium*, transgenes, and foreign genetic materials. Hence, there is a high probability that current GMO regulations could be avoided by CRISPR/Cas RNP-mediated precise genetic engineering [27,28]. However, the EU imposed more stringent regulations on GM foods than the US. The EU’s regulatory approach focuses more on the process than the product. As all GM foods are made with different processes than natural (conventional) foods, they are supposed to be regulated. All GM food products must require premarket approval and proper labeling.

Despite all of these, a good number of bioengineered crops have been commercialized [23,24]. Bioengineered papaya, sweet corn, squash, potato, apple, and eggplant have been released for fresh consumption [25]. Despite that, there have not been any safety issues noticed so far, but there is still concern about consumer acceptance of bioengineered products. The primary concern is somehow justified due to the non-directional changes resulting from genetic transformation in some cases [148]. Interestingly, there should be a different approach to mitigate or eliminate these off-target effects [115,120,121].

## 7. Conclusions

Tissue and developmental stage-specific mutagenesis of candidate gene(s) by CRISPR-TSKO and CRISPR-based precise nucleotide editing are favorable options for seedlessness breeding since these strategies overcome the interspecies hybridization barrier and prevent/minimize the pleiotropic effects of genetic engineering. Moreover, it could develop transgene-free if RNP-mediated transformation is subjected. Appropriate gene targeting, gRNA designing, appropriate promoter and Cas protein selection, cloning of all modules into a proper binary vector, transforming plant cells, and subsequent regeneration are the steps for genetic engineering-mediated seedlessness trait gaining. Finding or establishing a good transformation and plant regeneration system could largely improve the desired plant recovery rate. This article summarizes the genes that could be targeted with the CRISPR/Cas platform. It also proposed a model and provided sources of related, useful information for executing genome-editing projects for gaining seedlessness. This approach could be applied to other crops as well. However, in the case of working with plants of different ploidy, hybrids, and plants with different degrees of sex expression, additional technologies will be required for a preliminary assessment of the genotype and a special protocol for evaluating the results since it is not enough to guarantee them simply by the presence or absence of an edited genome region.

## Figures and Tables

**Figure 1 ijms-24-05660-f001:**
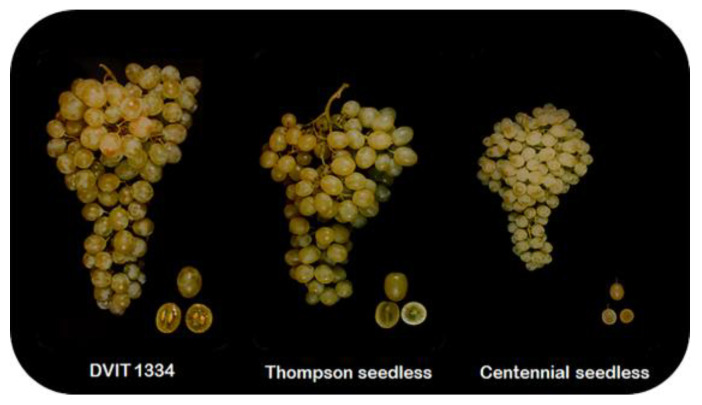
Close-up cluster views of Thompson seeded mutant (DVIT 1334), Thompson seedless, and Centennial seedless grape genotypes exhibiting seeded, stenospermocarpy, and parthenocarpy fruit set programs, respectively.

**Figure 2 ijms-24-05660-f002:**
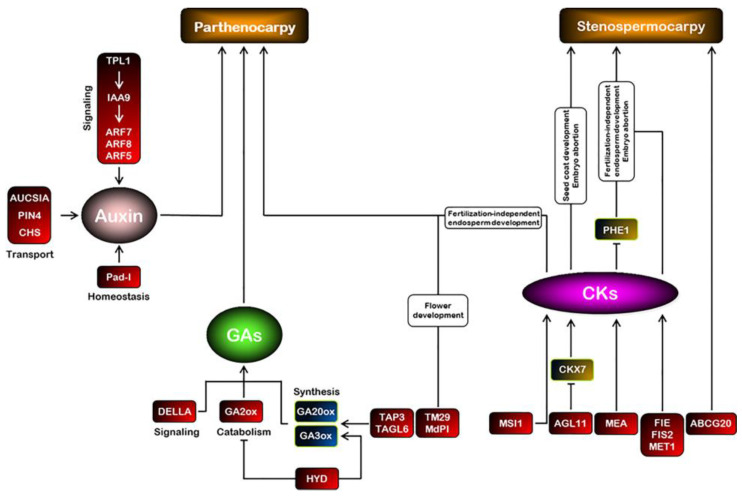
Schematic model of hormonal regulation of seedless fruit set. Parthenocarpy is obtained either by exogenous treatments or by genetic manipulations of phytohormones. Gene names in red boxes represent the gene loss-of-function mutation or downregulation that causes parthenocarpy or stenospermocarpy seedlessness. The gene name in the red boxes could be identified in Table 1. *GA20ox* (GA 20 oxidase) and *GA3ox* (GA 3 oxidase), GA biosynthetic genes; *GA2ox* (GA 2 oxidase), a GA catabolic enzyme; CKX7 (cytokinin oxidases/dehydrogenases), a CK-degrading enzyme; and *PHE1* (PHERES1), a type I MADS-box gene.

**Figure 3 ijms-24-05660-f003:**
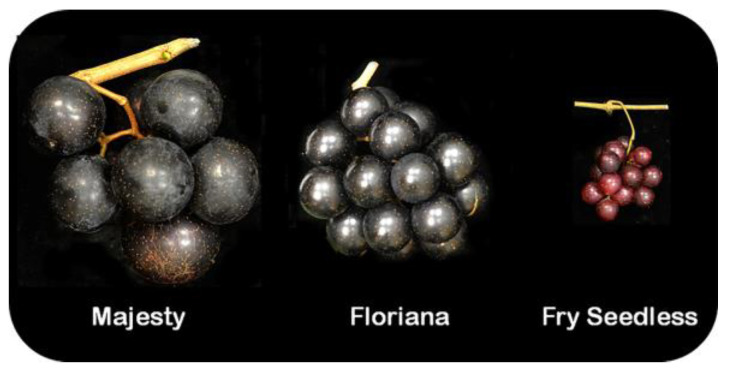
Close-up cluster view of muscadine cultivars Majesty (female flower, seeded), Noble (perfect flower, seeded), and Fry seedless (perfect flower, parthenocarpy seedless).

**Figure 4 ijms-24-05660-f004:**
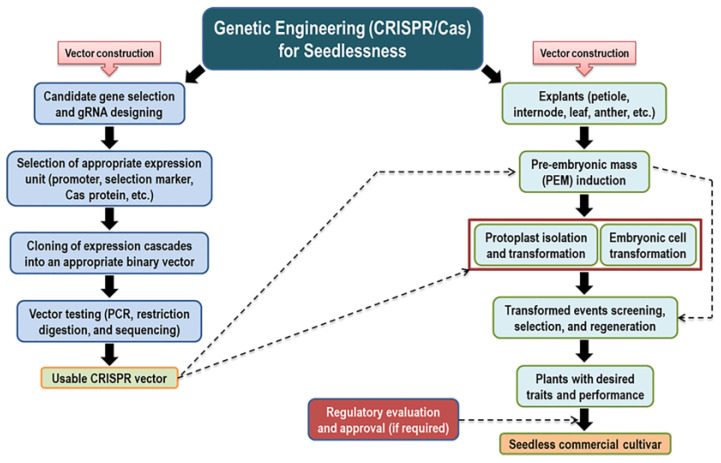
Visual scheme of gaining the seedlessness trait using genetic engineering (CRISPR/Cas) strategy.

## Data Availability

Not applicable.

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
