# Peer review of "Seedlessness Trait and Genome Editing—A Review"

_ijms, 2023, doi:10.3390/ijms24065660_

Round 1

Reviewer 1 Report

This paper provides an informative overview of the current state of research on seedlessness trait and genome editing. It discusses both the potential benefits and challenges associated with this technology, as well as existing methods for producing seedless fruits.

This paper is an important contribution to our understanding of this topic and should be considered by anyone interested in exploring further research in this area.

The paper is well-written and provides a comprehensive overview of the current research in this field. The authors provide a clear explanation of the various techniques used to achieve seedlessness, as well as their potential benefits and drawbacks.

However, there are some areas where the paper could be improved. For example, there is no discussion of how these techniques might be used in practice or what kind of regulatory framework might be necessary to ensure their safe use.

Additionally, there is no discussion of how these techniques might influence other aspects of agriculture, such as crop yields or pest control strategies. There is also no discussion of how these techniques might affect consumer preferences or public opinion about genetically modified foods.

Overall, this paper clearly explains the various techniques used to achieve seedlessness and their potential benefits and drawbacks. However, it could be improved by providing more information on how these techniques might be used in practice and what kind of regulatory framework might be necessary for their safe use. Additionally, it could benefit from further discussion on how these techniques might impact other aspects of agriculture and consumer preferences about genetically modified foods.

In conclusion, genome editing technology has enabled researchers to introduce desirable traits such as seedlessness trait into plants with greater precision than ever before. This technology has numerous potential applications in agriculture and food production but also raises important ethical considerations that must be taken into account when using it. Further research is needed in order to better understand the implications of this technology before it can be safely used on a large-scale basis in food production systems around the world.

Author Response

There are some areas where the paper could be improved. For example, there is no discussion of how these techniques might be used in practice or what kind of regulatory framework might be necessary to ensure their safe use. Additionally, there is no discussion of how these techniques might influence other aspects of agriculture, such as crop yields or pest control strategies. There is also no discussion of how these techniques might affect consumer preferences or public opinion about genetically modified foods.

Response: Thanks for the efforts and critical revision of our manuscript. In response to your comment, we have added a “visual scheme” and a description of how this technology can be used in practice. Please see Figure 4 and the description (lines 519-581).

We have added another section, “Bio-engineered food regulation and consumer acceptance”, in response to regulatory activities to safely use bio-engineered food (lines 582-626). Within this section, we added a short paragraph (lines 583-599) to address how these techniques might influence other aspects of agriculture. The main purpose of this article is to use genome-editing technology to gain fruit seedlessness. Though, it is true that the same technology can be used to improve other traits.

Reviewer 2 Report

The presented review article «Seedlessness Trait and Genome Editing - a Review» summarizes and systematizes modern data on the various mechanisms and genes involved for producing seedless fruit as well as CRISPR/Cas-mediated genome editing approaches and their improvements. The article is well written and systematized into separate subsections. The authors analyzed a large number of modern studies, which is confirmed by reference list. In my opinion, the authors have presented a high-quality review article that will be very useful to the world scientific community.

As a recommendation, I would suggest that the author supplement the review manuscript with a small section. There is an article in which parthenocarpy fruits were obtained as a result of genetic transformation (Agrobacterium- or ballistic-mediated transformation), not associated with the target gene directly involved in parthenocarpy. For example, numerous data are presented on the production of tomato seedless fruits among T0 transgenic plants. Additionally, Parthenocarpy and fruit malformations are common among independent transgenic tomato lines, expressing genes encoding different PR-proteins and antimicrobial peptides (DOI: 10.1134/S1062360414010044; DOI: 10.1007/s00709-018-1252-y). I invite the authors to complete and discuss non-directional changes resulting from genetic transformation in the formation of normal and abnormal parthenocarpy or seedless fruits. In my opinion, this small addition will greatly enhance the value of the review article.

Once again, I would like to thank the authors for the interesting and necessary review.

Author Response

I suggest that the author supplement the review manuscript with a small section. There is an article in which parthenocarpy fruits were obtained as a result of genetic transformation (Agrobacterium- or ballistic-mediated transformation), not associated with the target gene directly involved in parthenocarpy. For example, numerous data are presented on the production of tomato seedless fruits among T0 transgenic plants. Additionally, Parthenocarpy and fruit malformations are common among independent transgenic tomato lines, expressing genes encoding different PR-proteins and antimicrobial peptides (DOI: 10.1134/S1062360414010044; DOI: 10.1007/s00709-018-1252-y). I invite the authors to complete and discuss non-directional changes resulting from the genetic transformation in the formation of normal and abnormal parthenocarpy or seedless fruits. In my opinion, this small addition will greatly enhance the value of the review article.

Response:

We appreciate the revision quality and valuable comments of the reviewer. With due respect, the idea of this manuscript is to accumulate only the master genes positively involved in the regulation of ovule/seed development. So that these genes can be used in CRISPR/Cas knockout platform to deactivate them. Accordingly, the seed formation program could be disturbed, resulting in a seedless phenotype. Of course, there are numerous scientific reports discussing the discovery of an excellent set of genes that can induce seedless trait, but through overexpression. This type of genes cannot be employed based on the strategy fundamentals.

We also have added another section, “Bio-engineered food regulation and consumer acceptance”. We hope that will satisfy your concern about “non-directional changes resulting from genetic transformation” (lines 582-626).

Reviewer 3 Report

Seedlessness Trait and Genome Editing - a Review by

M. Moniruzzaman, Ahmed G. Darwish, Ahmed Ismail, Ashraf El-Kereamy, Violeta Tsolova, Islam El-Sharkawy is an analysis of own and literature data related to the control of the seedless trait in relation to the size of the fruit, as a valuable quality. It should be noted that the article reflects some promising technologies for solving this problem, for example, the CRISPR/Cas genome modification technology. In general, this reflects the modern trend of moving away from "transgenic and terrible GMOs" in favor of the consumer. From this perspective, it is justified. However, then the authors should change the title of the publication and mention the approach they hope to take so that consumers remain "in the dark about the violation of celibacy" and believe that these grapes are truly natural.

I leave such a logical trick on the conscience of the authors, but you should not be shy and indicate your position, it will even find many fans.

Even if this issue is not considered with a reasonable variant of real changes in the genome and the introduction of heterologous sequences encoding metabolic pathways and developmental processes with different methods of seed induction, questions of a purely theoretical nature remain for the article. These issues are related to the botanical and physiological aspects of the development of various economically valuable hybrids and grape lines. It is equally important to understand for which varieties such features are more relevant. Indeed, this feature is extremely interesting for table varieties, as well as for a number of varieties from which raw materials for wine and confiture are obtained.

The article should also analyze the developmental features of grapes during the formation of dioeciousness and monoeciousness, the factors of influence of pretreatments with auxins and heberilins, which cannot be ignored taking into account the analysis of gene expression and especially inheritance, so as to understand how these modifications will be inherited even during vegetative propagation after CRISPR/ Cas is not obvious.

Nevertheless, I think that the article can be reflected and published in the journal as promising and interesting.

I suggest authors:

1) change the title of the article

2) offer a visual scheme of the considered technological approaches

3) expand the aspect of applying other genetic engineering approaches and classical breeding approaches

4) consider the botanical, anatomical aspects of the development of fruits in varieties, forms and lines that differ in the severity of the sex, taking into account the hybrid origin.

The work can be accepted after entering these data.

Author Response

Question 1:

The authors should change the title of the publication and mention the approach they hope to take so that consumers remain "in the dark about the violation of celibacy" and believe that these grapes are truly natural.

I leave such a logical trick on the conscience of the authors, but you should not be shy and indicate your position, it will even find many fans.

Response:

We sincerely appreciate the reviewer's efforts and comments. With due respect, the idea of addressing consumers' concerns is very significant; however, we cannot highlight it meaningfully. In fact, this issue was not the focus of this article, and we strongly think that the discussion of this issue needs other types of backgrounds and expertise (i.e., consumer behavior, marketing, economic development, etc.) that, unfortunately, we lack them.

Our main focus in this article is to discuss the technical aspect of genetic engineering mediated gaining of seedless trait, though we talked about the regulatory processes of the bio-engineered product as it has some relation with bringing the product to light. We shortly discuss the window of skipping the long-awaited regulatory evaluation if a certain technique is being followed.

To make it more obvious and satisfy the reviewer, we added another section, “Bio-engineered food regulation and consumer acceptance”, in the current version (lines 582-626).

Question 2:

Even if this issue is not considered with a reasonable variant of real changes in the genome and the introduction of heterologous sequences encoding metabolic pathways and developmental processes with different methods of seed induction, questions of a purely theoretical nature remain for the article. These issues are related to the botanical and physiological aspects of the development of various economically valuable hybrids and grape lines. It is equally important to understand for which varieties such features are more relevant. Indeed, this feature is extremely interesting for table varieties, as well as for a number of varieties from which raw materials for wine and confiture are obtained.

Response:

In the current manuscript, we only highlighted genes with functional experimental validation rather than population genetics, transcriptomic data, differential expression, or solely theoretical. Regarding the concern of “targeted variety of being improved”, We agree with the reviewer's opinion. However, we strongly believe that this is the homework of the scientists to know the genetic makeup of their own material to be used.

Question 3:

The article should also analyze the developmental features of grapes during the formation of dioeciousness and monoeciousness, the factors of influence of pretreatments with auxins and heberilins, which cannot be ignored taking into account the analysis of gene expression and especially inheritance, so as to understand how these modifications will be inherited even during vegetative propagation after CRISPR/ Cas is not obvious.

Response:

Regarding the concern of inherited traits, it could be achieved by applying the CRISPR reagent (RNP) to the germline cells using viral vectors like the Tobacco rattle virus (TRV). We have mentioned it in the section “Genetic engineering strategies for seedlessness breeding”.

Question 4:

Change the title of the article

Response: In this manuscript, we focused on the applicable genes related to the target trait and the technological aspects, though we honor customer choice and acceptance. However, the manuscript does not really focus on this issue. Thank you for your understanding.

Question 5:

Offer a visual scheme of the considered technological approaches

Response:

We have added a visual scheme and discussed it in the section “Genetic engineering strategies for seedlessness breeding” (Figure 4).

Question 6:

Expand the aspect of applying other genetic engineering approaches and classical breeding approaches

Response:

Thank you for the reviewer's suggestion. In fact, the manuscript highlights the new smart breeding technology as a strategy where using other breeding technologies is incapable to introduce the trait (lines 90-93). Our target was how we could gain seedlessness trait using CRISPR/Cas genome editing technique, as it is precise and updated to overcome the limitations of classical breeding. A statement with this meaning has been added (lines 583-599) to satisfy the reviewer's suggestion.

Question 7:

Consider the botanical, anatomical aspects of the development of fruits in varieties, forms and lines that differ in the severity of the sex, taking into account the hybrid origin.

Response:

Thanks a lot for the valuable suggestion. However, the botanical and anatomical aspects of the development of fruits and the severity of the sex are very wide subjects. We strongly think that adding this sub-title will considerably extend the manuscript and make the reader out of focus.

Round 2

Reviewer 3 Report

The authors of the manuscript have made significant changes to the manuscript and it can be accepted.

However, I recommend moving the link conclusion section to a discussion section. And also indicate that in the case of working with plants of different ploidy, hybrids and plants with different degrees of sex expression, additional technologies will be required for a preliminary assessment of the genotype and a special protocol for evaluating the results, since it is not enough to guarantee them simply by the presence of the absence of an edited genome region.

Author Response

Reviwer-3

I recommend moving the link conclusion section to a discussion section. And also indicate that in the case of working with plants of different ploidy, hybrids and plants with different degrees of sex expression, additional technologies will be required for a preliminary assessment of the genotype and a special protocol for evaluating the results, since it is not enough to guarantee them simply by the presence of the absence of an edited genome region.

Response:

Thanks a lot for the positive feedback. We removed the link and related sentences and adjusted it into subsection “Selecting appropriate candidate gene(s) and regulatory elements for targeted genome editing” of the main section “Genetic engineering strategies for seedlessness breeding”.

In the “conclusion”, we made the necessary modifications and added the referred sentences, as suggested by the reviewer.